# Impacts of Dietary Standardized Ileal Digestible Lysine to Net Energy Ratio on Lipid Metabolism in Finishing Pigs Fed High-Wheat Diets

**DOI:** 10.3390/ani14121824

**Published:** 2024-06-19

**Authors:** Jiguang Wang, Haojie Li, He Zhu, Shuangshuang Xia, Fang Zhang, Hui Zhang, Chunxue Liu, Weijiang Zheng, Wen Yao

**Affiliations:** 1College of Animal Science and Technology, Nanjing Agricultural University, Nanjing 210095, China; wangjiguang20@163.com (J.W.); 2021105068@stu.njau.edu.cn (H.L.); 17863871658@163.com (H.Z.); zhengweijiang@njau.edu.cn (W.Z.); 2Anyou Biotechnology Group Co., Ltd., Suzhou 215437, China; xiashuangshuang@anschina.cn (S.X.); zhangfang@anschina.cn (F.Z.); zhanghui@anschina.cn (H.Z.); cx_liu@anschina.cn (C.L.); 3Key Lab of Animal Physiology and Biochemistry, Nanjing Agricultural University, Ministry of Agriculture and Rural Affairs of the People’s Republic of China, Nanjing 210095, China

**Keywords:** finishing pigs, SID Lys:NE ratio, marbling score, lipid metabolism, microbiome, metabolomics

## Abstract

**Simple Summary:**

Body metabolism and colonic microbiota collectively controlled lipid metabolism in finishing pigs. Diminishing standardized ileal digestible lysine to net energy ratio in high-wheat diets may regulate lipid metabolism via AMP-activated protein kinase α1/sirtuin 1/peroxisome proliferator-activated receptor-γ coactivator-1α pathway and farnesol X receptor/small heterodimer partner pathway, ultimately increased the marbling score of longissimus dorsi muscle.

**Abstract:**

The present study aimed to investigate the impacts of dietary standardized ileal digestible lysine to net energy (SID Lys:NE) ratio on lipid metabolism in pigs fed high-wheat diets. Thirty-six crossbred growing barrows (65.20 ± 0.38 kg) were blocked into two treatment groups, fed high-wheat diets with either a high SID Lys:NE ratio (HR) or a low SID Lys:NE ratio (LR). Each treatment group consisted of three replicates, with six pigs per pen in each replicate. The diminishing dietary SID Lys:NE ratio exhibited no adverse impacts on the carcass trait (*p* > 0.05) but increased the marbling score of the longissimus dorsi muscle (*p* < 0.05). Meanwhile, LR diets tended to increase the serum triglyceride concentration (*p* < 0.1). LR diets upregulated fatty acid transport protein 4 and acetyl-coA carboxylase α expression levels and downregulated the expression level of adipose triglyceride lipase (*p* < 0.05). LR diets improved energy metabolism via decreasing the expression levels of AMP-activated protein kinase (AMPK) α1, sirtuin 1 (SIRT1), and peroxisome proliferator-activated receptor-γ coactivator-1α (PGC-1α) (*p* < 0.05). Additionally, LR diets stimulated hepatic bile acid synthesis via upregulating the expression levels of cytochrome P450 family 7 subfamily A member 1 and cytochrome P450 family 27 subfamily A member 1, and downregulating farnesol X receptor (FXR) and small heterodimer partner (SHP) expression levels (*p* < 0.05). A lowered SID Lys:NE ratio affected the colonic microbial composition, characterized by increased relative abundances of *YRC22*, *Parabacteroides*, *Sphaerochaeta*, and *Bacteroides*, alongside a decreased in the proportion of *Roseburia*, *f_Lachnospiraceae_g_Clostridium*, *Enterococcus*, *Shuttleworthia*, *Exiguobacterium*, *Corynebacterium*, *Subdoligranulum*, *Sulfurospirillum*, and *Marinobacter* (*p* < 0.05). The alterations in microbial composition were accompanied by a decrease in colonic butyrate concentration (*p* < 0.1). The metabolomic analysis revealed that LR diets affected primary bile acid synthesis and AMPK signaling pathway (*p* < 0.05). And the mantel analysis indicated that *Parabacteroides*, *Sphaerochaeta*, *f_Lachnospiraceae_g_Clostridium*, *Shuttleworthia*, and *Marinobacter* contributed to the alterations in body metabolism. A reduced dietary SID Lys:NE ratio improves energy metabolism, stimulates lipogenesis, and inhibits lipolysis in finishing pigs by regulating the AMPKα/SIRT1/PGC-1α pathway and the FXR/SHP pathway. *Parabacteroides* and *Sphaerochaeta* benefited bile acids synthesis, whereas *f_Lachnospiraceae_g_Clostridium*, *Shuttleworthia*, and *Marinobacter* may contribute to the activation of the AMPK signaling pathway. Overall, body metabolism and colonic microbiota collectively controlled the lipid metabolism in finishing pigs.

## 1. Introduction

China, the world’s largest pork producer and consumer, produced 671.28 million pigs in 2021 according to the National Bureau of Statistics [1]. Most of swine husbandry depends on corn–soybean meal diets, the increasing demand for corn and soybean directly threatens the China’s food security. In 2021, the Ministry of Agriculture and Rural Affairs of the People’s Republic of China has issued an annual plan for reducing and replacing corn and soybean meal, in which 10–20% of wheat can be incorporated into growing and finishing pig formula (http://www.moa.gov.cn/gk/nszd_1/2021/202104/t20210421_6366304.htm, accessed on 15 February 2024). As a vital livestock feed source, wheat is increasingly being included in pig diets [2,3]. Han et al. [3] found that growing–finishing pigs can include up to 60% wheat in their feed without detrimental impact on growth. Wheat contains higher levels of crude protein and concentrations of amino acids (AAs), like lysine, tryptophan, and threonine, than corn [3]. And wheat can provide up to 60% of the pig’s requirement for total AAs [4] and up to 70% for essential AAs [5]. Undoubtedly, employing wheat in pig diets will effectively decrease the dependency on corn and soybean, ensuring a stable and secure supply of food.

Intramuscular fat (IMF) content, represented by the marbling score, stands as one of the most crucial indicators for grading pork quality [6]. Bejerholm and Barton-Gade proposed a minimum of 2% IMF to ensure satisfactory eating characteristics [7], and DeVol et al. reported a linear preference for IMF in terms of juiciness, tenderness, and a flavor range from 1.09% to 6.75% [8]. However, wheat has a lower digestible energy and metabolic energy than corn when fed to barrows [9]. Feeding high-wheat diets (with wheat content at 55%) or wheat–soybean meal diets (replacing corn with wheat entirely) can compromise pork quality, such as IMF content [10,11]. Mastering the intricate mechanism behind wheat-induced meat quality deterioration and resolving this challenge as a pivotal endeavor in the realm of scientific inquiry. The previous research in our group found that high-wheat diets inhibited lipogenesis, stimulated lipolysis, and ultimately reduced IMF content (unpublished data). Host low energy states and colonic short-chain fatty acids (SCFAs)-producing bacteria alteration were responsible for the reduction in IMF content [12]. Therefore, improving energy metabolism and gut microbiota is key to alleviating the impairment in meat quality resulting from high-wheat diets. It has been confirmed consuming corn–soybean meal diets with a low lysine to net energy ratio leads to an enhancement in IMF content in pigs via improving host energy metabolism [13]. Simultaneously, dietary lysine to energy ratio can affect gut microbiota [14]. Nevertheless, its impact on lipid metabolism disorder in finishing pigs resulting from wheat consumption remains uncertain. We hypothesized that a diminishing dietary standardized ileal digestible lysine to net energy (SID Lys:NE) ratio may promote IMF accumulation via regulating energy metabolism and intestinal microbiota in pigs fed high-wheat diets. This study aimed to study the impacts of dietary SID Lys:NE ratio on the carcass, body metabolism and intestinal microbiota, as well as its correlation with lipid metabolism in pigs fed high-wheat diets using combined microbiome–metabolomic approaches.

## 2. Materials and Methods

### 2.1. Animal and Experimental Design

Thirty-six crossbred (Duroc × Landrace × Yorkshire) growing barrows (65.20 ± 0.38 kg) were blocked into two treatment groups, fed high-wheat diets with either a high SID Lys:NE ratio (HR) or a low SID Lys:NE ratio (LR). Each treatment group consisted of three replicates, with six pigs per pen in each replicate. Dietary wheat level (55%) was established based on our prior unpublished data, revealing a noteworthy decline in IMF content upon reaching this level. The feeding trial was divided into two phases (65–90 kg and 90–110 kg), the pigs were given unrestricted access to feed and water. Pigs were housed in an environmentally controlled building with pens equipped with a feeder, a nipple drinker, and slotted floors during the experiment period. The feeding management and immunization program were conducted according to the general requirements of the pig farm. The composition and nutrient levels of the experimental diets were given in Table 1. The experiment was approved by the Animal Care and Use Committee of Nanjing Agricultural University (Approve number SYXK (Su) 2011-0036).

### 2.2. Sample Collection

When body weight (BW) reached market weight, twelve pigs with BW ranging 107.04 to 127.10 kg (two pigs with medium BW from each replicate) were chosen for evaluation. After an overnight fast, urine samples were collected from chosen pigs. Pigs were restrained by snout snare, and blood samples were collected from the anterior vena cave with a 14-gauge needle. Then, serum samples were obtained after centrifugation at 3000 rpm for 15 min at 4 °C. Pigs were slaughtered by exsanguination after euthanasia by electrical stunning (head only, 110 V, 60 Hz). After opening the hot carcasses along the ventral midline, the liver samples were taken from the left liver lobe and the colonic chyme was collected from the proximal colon. Liver and colonic chyme samples were frozen in liquid nitrogen and stored at −80 °C for subsequent analysis.

### 2.3. Collection of Carcass Data and Marbling Score of Longissimus Dorsi Muscle (LD)

The hot carcass weight (HCW) was recorded within 30 min of slaughter and used for calculating the dressing percentage. Backfat thickness was measured at the thoracolumbar junction, the lumbosacral junction, and the thickest position on the shoulder using a vernier caliper (Shanghai Meinaite Industrial Co., Ltd., Shanghai, China) and the average value was defined as average backfat thickness. The loin-eye area (LEA) on the left side between the 10th and 11th ribs was determined by a QCJ-2A digital planimeter (Harbin Optical Instrument Factory Co. Ltd., Heilongjiang, China) and the LEA was calculated as the average of three measurements. LD at last rib was removed from the left side, and the marbling score of LD was calculated according to the NPPC Guidelines [15].

### 2.4. Serum Biochemical Parameters Assays

The concentrations of glucose (GLU), total cholesterol (TC), triglyceride (TG), high density lipoprotein cholesterol (HDL-C), low density lipoprotein cholesterol (LDL-C), and non-esterified fatty acid (NEFA) in serum were detected using commercial kits (Medicalsystem Biotechnology Co., Ltd., Zhejiang, China) with a HITACHI 7020 biochemistry analyzer (HITACHI Ltd., Tokyo, Japan).

### 2.5. Quantitative Real-Time PCR

Total RNA was obtained from liver samples using the RNAiso Plus reagent (Takara, Dalian, China) and reverse-transcribed to cDNA with ABScript III RT Master mix for qPCR with gDNA Remover (RK20429, ABclonal, Wuhan, China). The quantification of target gene mRNA expression levels was conducted using real-time PCR, following the methodology described by Wang et al. [11]. The 2^−ΔΔCt^ method, with normalization to the glyceraldehyde-3-phosphate dehydrogenase (GAPDH) gene, was employed to calculate the mRNA expression level of target genes. The primers sequences for AMP-activated protein kinase α1 (AMPKα1), sirtuin 1 (SIRT1), peroxisome proliferator-activated receptor-γ coactivator-1α (PGC-1α), farnesol X receptor (FXR), small heterodimer partner (SHP), cytochrome P450 family 7 subfamily A member 1 (CYP7A1), cytochrome P450 family 27 subfamily A member 1 (CYP27A1), cytochrome P450 family 8 subfamily B member 1 (CYP8B1), cytochrome P450 family 7 subfamily B member 1 (CYP7B1), fatty acid transport protein 4 (FABP4), fatty acid translocase (CD36), stearoyl-CoA desaturase (SCD), acetyl-coA carboxylase α (ACCα), acetyl-coA carboxylase β (ACCβ), fatty acid transport protein 2 (FATP2), hormone-sensitive lipase (HSL), adipose triglyceride lipase (ATGL), carnitine palmitoyl transferase (CPT), and GAPDH are given in Appendix A.

### 2.6. DNA Extraction, MiSeq Sequencing, and Bioinformation Analysis

The total DNA from colonic chyme were extracted with QIAamp Fast DNA Stool Mini Kit (QIAGEN, Hilden, Germany). A forward primer 338F (5′-ACTCCTACGGGAGGCAGCA-3′) and a reverse primer 806R (5′-GGACTACHVGGGTWTCTAAT-3′) were used to amplify the V3–V4 regions of bacterial 16S rRNA gene. Purified amplicons were pooled in equimolar and paired-end sequenced (2 × 250) on the Illumina Novaseq PE250 platform. Raw sequence was demultiplexed and quality filtered with Quantitative Insights Into Microbial Ecology 2 (QIIME2, version 2019.4). Then, sequences were denoised, merged and chimera removed using the Divisive Amplicon Denoising Algorithm 2, meanwhile amplicon sequence variants (ASVs) were constructed. A phylogeny was constructed using non-singleton ASVs with fasttree2. The phylogenetic affiliation of each ASVs was analyzed by the classify-sklearn naive Bayes taxonomy classifier.

For microbial community profiling, a Veen diagram was created to show the common and distinctive ASVs between groups. α-diversity (Chao1 richness estimator on the ASVs level) was assessed with a non-parametric factorial Kruskal–Walli’s test. The principal coordinates analysis (PCoA) ordination plots based on the Jaccard distance metric were employed to depict variations in β-diversity, and the difference were assessed by PERMANOVA. Then, linear discriminant analysis (LDA) effect size (LEfSe) with an alpha value of 0.5 Kruskal–Walli’s test was applied to identify all species with significant differential abundance. The LDA score (log10 LDA) threshold was established at 2.0.

### 2.7. Short-Chain Fatty Acids Concentration Analysis

The SCFAs concentrations in the colonic chyme were measured by using a capillary column gas chromatograph (GC-14B, Shimadzu, Japan; Capillary Column: 30 m × 0.32 mm × 0.25 μm film thickness) according to the method described in a previous study [16].

### 2.8. Metabolome Analysis

Urine samples were thawed at 4 °C and then vortexed for 60 s. Then, 100 μL samples were transferred into a 2.0 mL centrifuge tube and mixed with an equal volume of 2-Amino-3-(2-chloro-phenyl)-propionic acid (4 ppm) solution prepared with 80% methanol water. The mixed solution was centrifuged for 10 min at 12,000 rpm and 4 °C to obtain supernatant. And the supernatant was filtered with 0.22 μm membrane and transferred into the detection bottle for liquid chromatograph mass spectrometer (LC-MS) detection, which was performed by PANOMIX. The MzXML files, converted from raw MS data with MSConvert in the ProteoWizard software package (version 3.0.8789), were imported into XCMS software (version 3.12.0) for alignment, peak detection, and retention-time corrections. The metabolites were identified and cross-referenced with HMDB, Massbank, LipidMaps, mzcloud, and KEGG databases. The ClassyFire classification system was used to categorize metabolites. The differential metabolites were identified with discriminant analysis of orthogonal partial least squares (OPLS-DA) based on importance projection (VIP) values of 1.0 and *p*-value of 0.05. Differential metabolites were visualized by volcano and heatmap plots. Meanwhile, differential metabolites were subjected to pathway analysis using an online tool (http://www.metaboanalyst.ca/faces/ModuleView.xhtml, accessed on 23 January 2024).

### 2.9. Statistical Analysis

SPSS 25.0 (IBM Inc. Chicago, IL, USA) was used for data analyses of carcass traits, marbling score of LD, serum biochemical indexes, and gene expression levels with a Student’s *t*-test. The difference in colonic SCFAs concentration between groups were compared with non-parametric Mann–Whitney U test. Data (carcass traits, marbling score of LD, serum biochemical indexes, genes expression levels) were expressed as the means and their SEM. Differences were considered statistically significant at *p* < 0.05, and 0.05 ≤ *p* < 0.10 was considered as a trend towards significance. Correlations between the relative abundance of changed genera with concentration of SCFAs and differential metabolites were estimated by Spearman’s rank correlation. Additionally, mantel test was employed to detect the correlation between microbiota and urine metabolite profiles using the OmicStudio tools (https://www.omicstudio.cn/tool, accessed on 23 January 2024).

## 3. Results

### 3.1. Carcass Rraits and Marbling Score of LD

The effects of dietary treatment on carcass traits and marbling score of LD were presented in Table 2. The marbling score of LD was increased in pigs fed with LR diets compared to HR diets (*p* < 0.05). And the dietary SID Lys:NE ratio had no significant effects on pre-slaughter BW, HCW, dressing percentage, backfat thickness at thickest shoulders, thoracolumbar junction, lumbosacral junction, and average backfat thickness, as well as LEA (*p* > 0.05).

### 3.2. Serum Biochemical Indexes

Table 3 showed the effects of dietary SID Lys:NE ratio on serum biochemical indexes of finishing pigs. Diminishing dietary SID Lys:NE ratio showed a tendency to increase the serum TG concentration (*p* < 0.1), and no significant effects of dietary treatment on the concentrations of GLU, TC, HDL-C, LDL-C, and NEFA (*p* > 0.05) in serum were found.

### 3.3. The Energy Metabolism, Bile Acid Synthesis, and Lipid Metabolism-Related Gene Expression in Liver

In comparison with the HR group, the mRNA expression levels of AMPKα1, SIRT1 and PGC-1α were lower (*p* < 0.05) in the LR group (Figure 1A). LR diets downregulated FXR and SHP mRNA expression levels (*p* < 0.05), upregulated CYP7A1 and CYP27A1 mRNA expression levels (*p* < 0.05) and tended to decrease CYP7B1 mRNA expression level (Figure 1B) (*p* < 0.1). Compared with the HR groups, the pigs had higher expression levels of FABP4 and ACCα (*p* < 0.05), as well as a lower expression level of ATGL (*p* < 0.05) in the LR group (Figure 1C,D). Meanwhile, there was no significant difference in the mRNA expression levels of CYP8B1, FATP2, CD36, SCD, ACCβ, HSL, or CPT between groups (*p* > 0.05).

### 3.4. Colonic Microbial Community

To explore the effects of the dietary SID Lys:NE ratio on the colonic microbiome of finishing pigs, the microbial diversity and composition were compared. There were 7565 and 9370 unique ASVs in the HR and LR groups, respectively, with 5541 shared ASVs among the groups (Figure 2A). Dietary treatment had no significant effects on α-diversity (Figure 2B) (*p* > 0.05), and the PCoA plot did not reveal a separation of microbiota (Figure 2C) (PERMANOVA, *p* > 0.05). The major intestinal microbiota species in phylum were *Firmicutes* (75.38%), *Bacteroidetes* (21.89%), *Proteobacteria* (1.05%), and *Spirochaetes* (0.76%) (Figure 2D). Figure 2E presented the composition of the microbiota at the top 30 genus level, the 10 major classified genera among groups were *Prevotella* (18.21%), *SMB53* (10.03%), *Streptococcus* (4.52%), *Lactobacillus* (6.28%), *Roseburia* (3.34%), *Blautia* (3.30%), *f_Clostridiacea_g_Closttridium* (1.42%), *Faecalibacterium* (0.60%), *Coprococcus* (0.83%), and *Megasphaera* (0.73%). The LEfSe analysis identified bacteria as biomarkers for microbiota from phylum to genus (Figure 3A). At the genus level, *YRC22*, *Parabacteroides*, *Sphaerochaeta*, and *Bacteroides* became enriched in the LR group, while *Roseburia*, *f_Lachnospiraceae_g_Clostridium*, *Enterococcus*, *Shuttleworthia*, *Exiguobacterium*, *Corynebacterium*, *Subdoligranulum*, *Sulfurospirillum*, and *Marinobacter* were markedly higher in the HR group (Figure 3B–N) (*p* < 0.05).

### 3.5. SCFAs Concentrations and Their Correlation with Differential Genera

As indicated in Figure 4A, diminishing dietary SID Lys:NE ratio tended to reduce colonic butyrate concentration (*p* < 0.1). And the heatmap suggested that the relative abundance of differential genera had significant correlations with the concentration of SCFAs in colon. For instance, the relative abundances of *Roseburia*, *f_Lachnospiraceae_g_Clostridium*, *Shuttleworthia*, *Corynebacterium*, and *Subdoligranulum* had positive correlations with the concentration of butyrate, whereas the proportion of *YRC22* and *Parabacteroides* had negative correlations with the concentration of butyrate (Figure 4B) (*p* < 0.05).

### 3.6. Nontargeted Metabolomic Analysis

The OPLS-DA scores plot showed separated clusters between the LR and HR groups, suggesting that lowering dietary SID Lys:NE ratio considerably altered metabolite profiles of urine in finishing pigs (Figure 5A). As shown in Figure 5B,C, a total of 751 metabolites were identified, of which 29 metabolites differed between groups (VIP > 1, *p* < 0.05). Compared to the HR group, levels of 14 metabolites were increased while 15 metabolites were decreased in the LR group. According to the ClassyFire classification system, metabolites were divided into 14 classes. And the differential metabolites comprised three benzene and substituted derivatives, five carboxylic acids and derivatives, two fatty acyls, two isoflavonoids, two organonitrogen compounds, two pyrimidine nucleosides, and so on. To explore the different metabolic pathways that respond to dietary SID Lys:NE ratio, the differential metabolites were imported into KEGG for enrichment analysis. The differential metabolites were involved in 34 metabolic pathways in total, with glutathione metabolism, vitamin digestion and absorption, acridone alkaloid biosynthesis, cholesterol metabolism, etc. pathways being affected by dietary SID Lys:NE ratio (Figure 5D).

### 3.7. Microbiota–Metabolites Correlation Analysis

As seen in Figure 6, there were significant correlations among differential metabolites. And mantel analysis indicated that twelve genera had powerful positive relationship with differential metabolites, including *Bacteroides*, *Clostridium*, *Corynebacterium*, *Enterococcus*, *Exiguobacterium*, *Marinobacter*, *Parabacteroides*, *Roseburia*, *Shuttleworthia*, *Sphaerochaeta*, *Subdoligranulum*, and *Sulfurospirillum* (r > 0.3, *p* < 0.05).

## 4. Discussion

The growth performance and carcass yield of finishing pigs directly influences the breeders’ profits from their production of pork. In our research, dietary SID Lys:NE ratio showed no significant effects on the growth performance of finishing pigs. As economical traits, carcass traits and marbling score of muscle not only directly determine the profits of producers, but also impact consumers’ desire to purchase the pork. In the present study, lowering the dietary SID Lys:NE ratio did not show adverse effects on dressing percentage, average fat thickness, or LEA. The marbling score, which has been proven to directly reflect IMF content [17], increased in the LR group. The finding offered compelling evidence supporting the notion that a reduction in the dietary SID Lys:NE ratio facilitated IMF deposition, as indicated by the increase in serum TG concentration. Hepatic TG synthesis and secretion were central to maintaining serum TG homeostasis [18]. The increase in the marbling score and serum TG level may indicate that LR diets altered host lipid metabolism, especially in the liver. Consequently, we examined the expression levels of lipid metabolism genes in the liver. The results indicated that pigs had higher expression levels of lipogenic genes (FABP4 and ACCα) and a lower expression level of ATGL in the LR group than in the HR group, suggesting that a diminishing dietary SID Lys:NE ratio triggered increased lipogenesis and inhibited lipolysis. Meanwhile, the expression level of AMPKα1, a key regulator of lipid metabolism through multiple mechanisms, was downregulated in the LR group. The AMPKα/SIRT1/PGC-1α pathway affects lipid metabolism via maintaining body energy homeostasis [19]. The downregulation of AMPKα1, SIRT1, and PGC-1α in the LR group hinted at the possibility that a diminishing dietary SID Lys:NE ratio improved energy metabolism. More energy was used for lipid synthesis, and energy production from fatty acids was inhibited. On the one hand, AMPKα1 directly regulated the expression of downstream lipid metabolism genes. The downregulation of AMPKα1 could upregulate the expression levels of FABP4 [19] and ACCα [20], while inhibited ATGL expression level [21]. ATGL is a key regulatory enzyme involved in lipolysis and participates in triglyceride-specific activity. On the other hand, the attenuation of AMPKα and SIRT1 suppressed β-oxidation of fatty acids by downregulating CPT expression [22], which transports long-chain fatty acids into mitochondria for energy production [23]. Although the CPT expression level was not different significantly among groups, we observed a numerical decrease in CPT expression level in the LR group, which indicated that LR diets inhibited fatty acids β-oxidation. Furthermore, the bile acids (BAs) pool size and composition played an essential role in the regulation of lipid homeostasis [24]. Cholesterol is converted to primary BAs via the classic pathway and alternative pathway. In the classic pathway, cholesterol is hydroxylated by CYP7A, converting it into 7α-hydroxycholesterol. Subsequently, 7α-hydroxycholesterol undergoes further modification into either cholic acid or chenodeoxycholic acid, catalyzed by CYP8B1 or CYP27A1, respectively. CYP7B1 is an important enzyme in the alternative pathway. In the alternative pathway, cholesterol is catalyzed by CYP27A1 to produce 27-hydroxy cholesterol, which is further catalyzed by CYP7B1 to produce chenodeoxycholic acid. As a bile acid receptor, hepatic FXR can modulate bile acid synthesis through target gene SHP [25]. Inhibition of FXR led to the downregulation of SHP, therefore facilitating BAs synthesis and the expression of ACC [26]. The downregulation of FXR, SHP and CYP7B1, as well as the upregulation of CYP7A1, CYP27A1, and ACCα in the LR group, indicated that diminishing dietary SID Lys:NE ratio strengthened classic BAs biosynthesis and lipogenesis via FXR/SHP pathway. Additionally, increased bile acid, resulting from LR diets, enhanced the coefficient of apparent total tract digestibility of ether extract (Appendix A), potentially elucidating one of reasons for the improvement in energy metabolism in pigs fed LR diets. The above results implied that lowering dietary SID Lys:NE ratio modified lipid metabolism via the regulation of the AMPKα/SIRT1/PGC-1α pathway and FXR/SHP pathway.

Gut microbes and their metabolites influenced host metabolism and energy homeostasis [27]. Although the overall microbial composition at the phylum level was not considerably different, diminishing dietary SID Lys:NE ratio significantly changed the relative abundance of certain taxa, including nine families and thirteen genera. Among these differential genera (relative abundance > 0.01%), Bacteroides resided within the distinctive bacterial genus of the LR group, while *Shuttleworthia*, *Exiguobacterium*, and *Corynebacterium* belonged to the unique bacterial genera of the HR group. Additionally, the relative abundances of *YRC22*, *Parabacteroides*, and *Sphaerochaeta* were increased in the LR group, whereas the proportions of *Roseburia*, *f_Lachnospiraceae_g_Clostridium*, Enterococcus, *Subdoligranulum*, *Sulfurospirillum*, and *Marinobacter* were decreased. Meanwhile, lowering the dietary SID Lys:NE ratio tended to reduce the colonic concentration of butyrate. An infusion of SCFAs (acetate, propionate, and butyrate) enhanced the mRNA expression level of CPT, while it inhibited fatty acid synthase mRNA expression in the liver of finishing pigs [28]. And butyrate has been substantiated for its ability to decrease lipid synthesis rates and induce fatty acid oxidation within the liver [29]. In our research, the butyrate content had significant positive correlations with the relative abundances of butyrate-producing bacteria *Roseburia* [30], *f_Lachnospiraceae_g_Clostridium* [31], *Shuttleworthia* [32], *Corynebacterium* [33], and *Subdoligranulum* [34]. We noted an inverse relationship between butyrate content and the proportion of *Parabacteroides*, which was consistent with findings from a prior study [35]. The LR group exhibited a notably lower relative abundance of the above butyrate producer compared to the HR group (1.64% vs. 5.42%), especially *Roseburia* (1.58% vs. 5.10%). Thus, we reckoned that a diminishing dietary SID Lys:NE ratio hindered butyrate production by reducing the relative abundance of butyrate-producing bacteria, mainly *Roseburia*. This, in turn, facilitated lipid synthesis and inhibited fatty acid oxidation. Subsequently, we explored the influence of dietary SID Lys:NE ratio on body metabolism using nontargeted metabolic analysis. The OPLS-DA model showed that the metabolite profiles were different between pigs in the two groups. The changes in lipids and lipid-like molecules, including 3-hydroxymethylglutaric acid, 5-acetamidovalerate, androsterone and 6-keto-prostaglandin F1α, suggested that dietary SID Lys:NE ratio influenced lipid metabolism in finishing pigs. Pathway enrichment analysis revealed that dietary SID Lys:NE ratio affected cholesterol metabolism and bile secretion, as evidenced by a significant increase in glycochenodeoxycholic acid (GCDCA) with a fold change of 44.18 (Appendix A, P = 0.015, VIP = 2.312). GCDCA is derived from chenodeoxycholic acid, therefore, the increased levels of GCDCA implied that the biosynthesis of chenodeoxycholic acid was stimulated in the liver. Moreover, LR diets decreased the concentration of 1,1-dimethylbiguanide (Appendix A, P = 0.041, VIP = 1.369), which was connected to the AMPK signaling pathway. The reduction in the concentration of 1,1-dimethylbiguanide, an AMPK activator [36], signified the suppression of the AMPK signaling pathway. The metabolomic analyses demonstrated that diminishing the dietary SID Lys:NE ratio promoted bile acid biosynthesis and suppressed the AMPK signaling pathway. These findings aligned with the results of gene expression analysis.

Microbiota–metabolites correlation analysis was performed to reveal the key genera manipulating body metabolism. In the present study, GCDCA and 1,1-dimethylbiguanide served as biomarkers for bile acid synthesis and AMPK signaling pathway alterations. Spearman’s correlation analysis showed that the concentration of GCDCA had positive correlation with the level of formylanthranilic acid. The mantel test demonstrated that significant positive correlations were observed between Parabacteroides and *Sphaerochaeta* with GCDCA and formylanthranilic acid. Moreover, the concentration of 1,1-dimethylbiguanide was positively correlated with the levels of 2-(Methylamino)benzoic acid and 5-acetamidovalerate. Although no correlation between differential genera with 1,1-dimethylbiguanide was observed, *f_Lachnospiraceae_g_Clostridium* and *Shuttleworthia* exhibited significant correlations with 2-(Methylamino)benzoic acid. Meanwhile, there was a significant correlation between *Marinobacter* with 5-acetamidovalerate. The above results implied that Parabacteroides and *Sphaerochaeta* benefited from BAs synthesis [37,38], whereas *f_Lachnospiraceae_g_Clostridium*, *Shuttleworthia*, and *Marinobacter* may contribute to the activation of AMPK signaling pathway. Taken as a whole, the microbiome played a vital role in the body metabolism alterations resulting from LR diets. In the first instance, the reduction in the relative abundance of butyrate-producing bacteria, coupled with a decrease in butyrate concentration, unequivocally stimulated lipogenesis, while inhibiting lipolysis. Alternatively, *Parabacteroides*, Sphaerochaeta, *f_Lachnospiraceae_g_Clostridium*, *Shuttleworthia*, and *Marinobacter* participated in the alterations in the BAs synthesis and the AMPK signaling pathway.

## 5. Conclusions

Dietary SID Lys:NE ratio exhibited no significant impacts on carcass trait. Diminishing dietary SID Lys:NE ratio increased the marbling score of LD and serum TG concentration. LR diets improved energy metabolism, stimulated lipogenesis and inhibited lipolysis by regulating the AMPKα/SIRT1/PGC-1α pathway and the FXR/SHP pathway. Changes in the gut microbial composition were responsible for the body metabolism alterations. *Parabacteroides*, *Sphaerochaeta*, *f_Lachnospiraceae_g_Clostridium*, *Shuttleworthia*, and *Marinobacter* participated in the alterations in the BAs synthesis and the AMPK signaling pathway. More importantly, the decline in the relative abundance of butyrate-producing bacteria and butyrate concentration contributed to the regulation in lipid metabolism (Figure 7).

## Figures and Tables

**Figure 1 animals-14-01824-f001:**
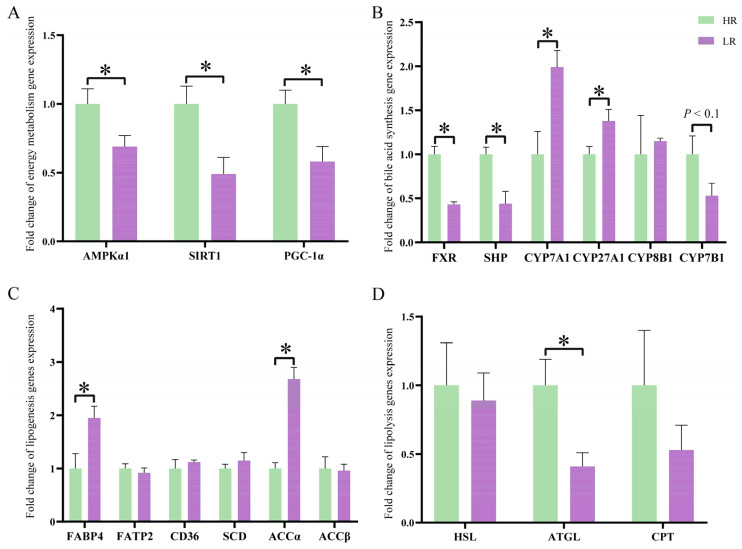
Effects of dietary standardized ileal digestible lysine to net energy ratio on hepatic gene expression levels of energy metabolism (**A**), bile acid synthesis (**B**), lipogenesis (**C**), and lipolysis (**D**). AMPKα1: AMP-activated protein kinase α1; SIRT1: sirtuin 1; PGC-1α: peroxisome proliferator-activated receptor-γ coactivator-1α; FXR: farnesol X receptor; SHP: small heterodimer partner; CYP7A1: cytochrome P450 family 7 subfamily A member 1; CYP27A1: cytochrome P450 family 27 subfamily A member 1; CYP8B1: cytochrome P450 family 8 subfamily B member 1; CYP7B1: cytochrome P450 family 7 subfamily B member 1; FABP4: fatty acid transport protein 4; CD36: fatty acid translocase; SCD: stearoyl-CoA desaturase; ACCα: acetyl-coA carboxylase α; ACCβ: acetyl-coA carboxylase β; FATP2: fatty acid transport protein 2; HSL: hormone-sensitive lipase; ATGL: adipose triglyceride lipase; CPT: carnitine palmitoyl transferase; GAPDH: glyceraldehyde-3-phosphate dehydrogenase. Data are expressed as the means ± SEM, * indicates significant difference (*p* < 0.05).

**Figure 2 animals-14-01824-f002:**
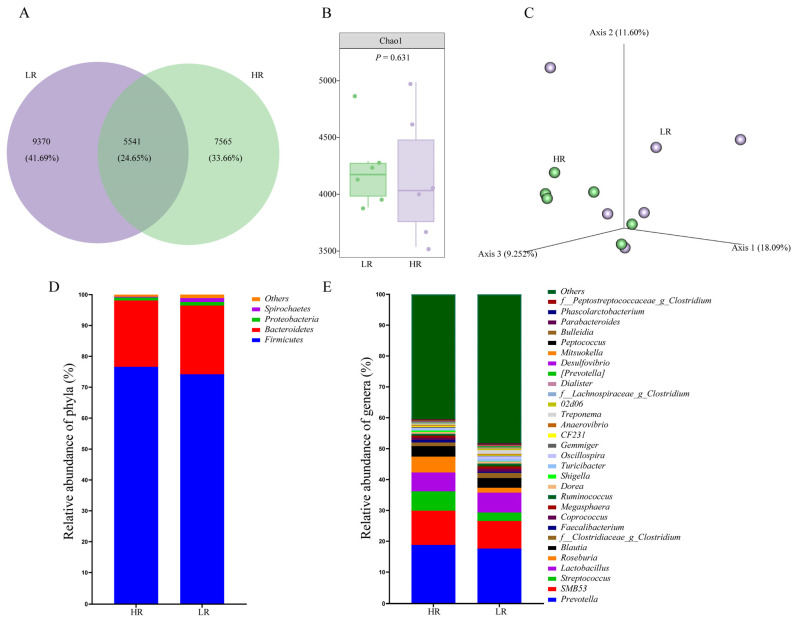
Effects of dietary standardized ileal digestible lysine to net energy ratio on the diversity and structure of colonic microbiota in finishing pigs. (**A**) A Veen diagram based on ASVs level. (**B**) Chao 1 index. (**C**) Principal coordinate analysis (PCoA) based on the Jaccard distance. (**D**) The top 4 dominant phyla and (**E**) top 30 genera abundance of bacteria.

**Figure 3 animals-14-01824-f003:**
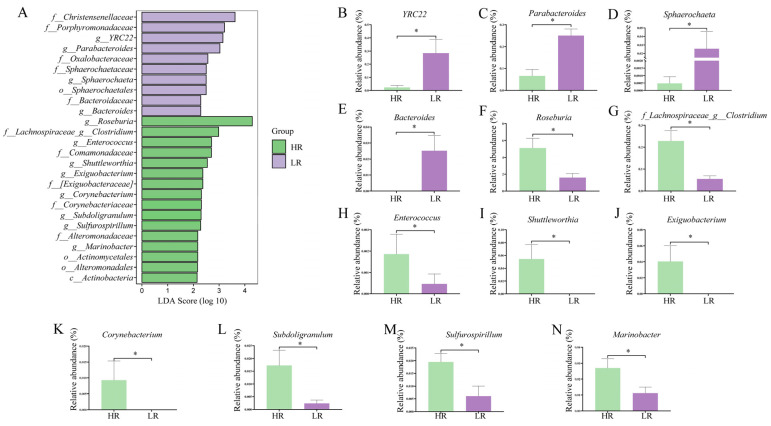
Effects of dietary standardized ileal digestible lysine to net energy ratio on the composition of colonic microbiota in finishing pigs. (**A**) Linear discriminant analysis effect size of microbiota. (**B**–**N**) The statistical analysis of genera. Data are expressed as the means ± SEM; * indicates significant difference (*p* < 0.05).

**Figure 4 animals-14-01824-f004:**
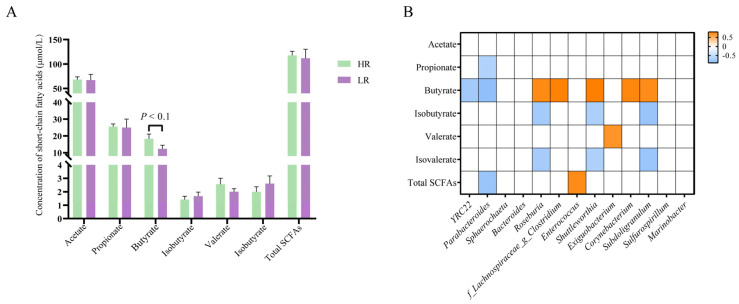
Effects of dietary standardized ileal digestible lysine to net energy ratio on the concentration of short-chain fatty acids (SCFAs) and the correlation of differential genera with SCFAs. (**A**) Concentration of SCFAs in colon. Data are expressed as the means ± SEM; * indicates significant difference (*p* < 0.05). (**B**) The correlation heatmap of differential genera with SCFAs. Cells are shaded with color exclusively when *p* < 0.05; the orange and blue colors represent positive and negative relationship, respectively.

**Figure 5 animals-14-01824-f005:**
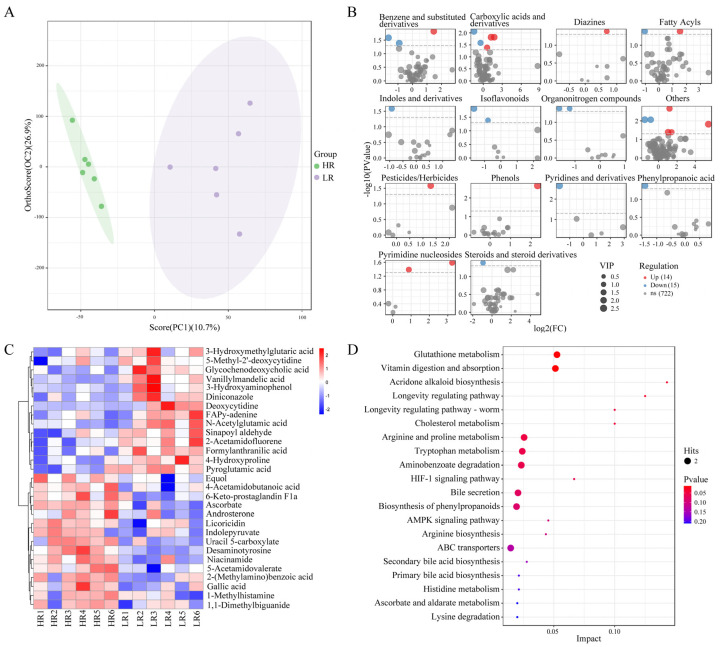
Effects of dietary standardized ileal digestible lysine to net energy ratio on urine metabolic profiles of finishing pigs. (**A**) Discriminant analysis of orthogonal partial least squares analysis (OPLS−DA) based on the urine compounds data. (**B**) The volcano plot of metabolites in LR vs. HR groups, metabolites from distinct classes were presented in separate graph. Red dots represent upregulated metabolites in the LR group, and the blue dots represent downregulated metabolites. (**C**) Heatmap of 29 differentially expressed metabolites: rows, metabolites; columns, samples. (**D**) Overview of pathway analysis based on metabolite alterations (top 20 pathways were shown).

**Figure 6 animals-14-01824-f006:**
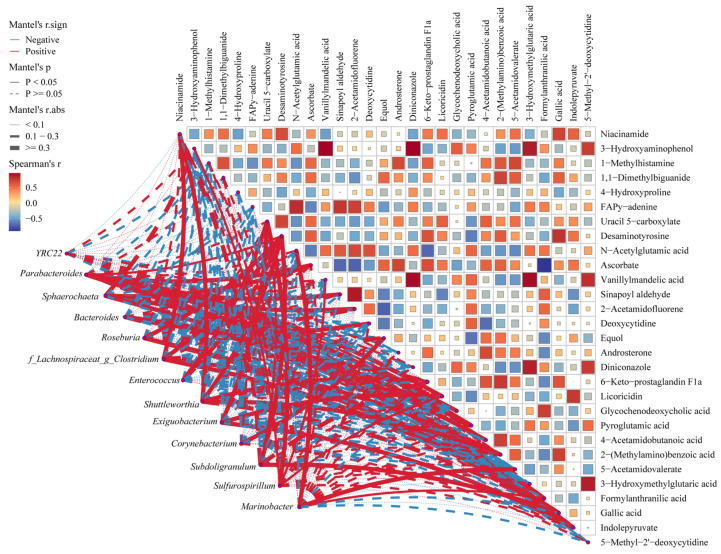
Microbiota–metabolites correlation. Pairwise comparisons of differential metabolites are demonstrated with a color gradient denoting Spearman’s correlation. Correlation between microbiota and metabolic profiles were analyzed with mantel test. Edge width corresponds to the Mantel’s r statistic for the corresponding distance correlation, and red color and bule color denote positive and negative correlation, respectively.

**Figure 7 animals-14-01824-f007:**
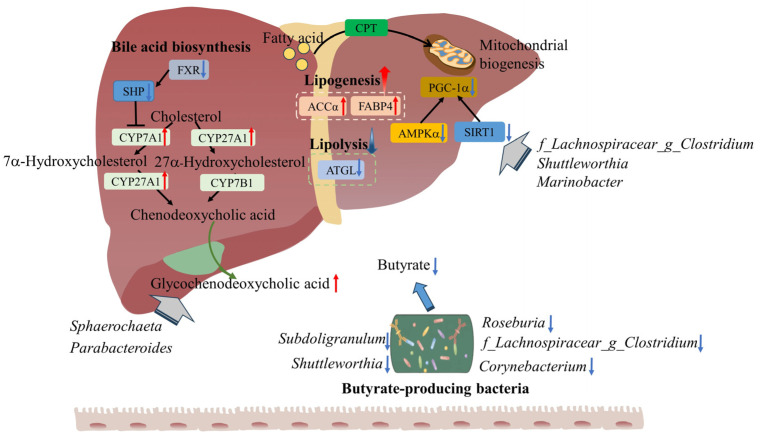
Potential mechanisms regulating lipid metabolism in finishing pigs fed diets with low standardized ileal digestible lysine to net energy ratio.

**Table 1 animals-14-01824-t001:** The composition and nutrient levels of diets for finishing pigs (as-fed basis).

Items	65–90 kg		90–110 kg	
	HR ^4^	LR ^5^	HR	LR
Ingredient, g				
Wheat	550.00	550.00	550.00	550.00
Corn	129.40	160.40	168.00	193.00
Soybean meal	110.00	75.00	54.00	33.00
Wheat middings	100.00	100.00	200.00	200.00
Rice bran	75.00	75.00	—	—
Limestone	11.00	8.60	10.00	8.50
Soybean oil	—	8.00	—	—
*L*-Lysine·HCl (70%)	5.80	5.45	5.20	4.50
*DL*-Methionine	0.95	0.65	0.55	0.20
*L*-Tryptophan	0.40	0.30	—	—
*L*-Threonine	1.80	1.60	1.60	1.30
Valine	0.35	0.25	—	—
NaCl	3.80	2.70	3.70	2.70
CaHCO_3_	3.70	4.20	—	—
NaHCO_3_	2.10	2.40	1.90	1.70
Choline chloride	0.50	0.50	0.20	0.20
CuSO_4_	0.30	0.30	0.30	0.30
Premix A ^1^	10.00	10.00	—	—
Premix B ^2^		—	10.00	10.00
Total	1005.10	1005.35	1005.45	1005.40
Nutrient composition ^3^, %				
Net energy (NE, MJ/kg)	9.75	10.08	9.62	9.96
Digestible energy (DE, MJ/kg)	13.73	13.95	13.61	13.63
Crude protein	16.07	14.74	14.44	13.63
Crude fat	3.01	3.88	2.33	2.39
Crude fiber	3.19	3.05	2.93	2.86
Standardized ileal digestible amino acids, %				
Lysine (Lys)	0.94	0.85	0.80	0.72
Methionine + Cysteine	0.56	0.50	0.49	0.44
Threonine	0.61	0.55	0.54	0.48
Tryptophan	0.18	0.16	0.15	0.14
Lys:NE (g/MJ)	0.96	0.84	0.83	0.72
Lys:DE (g/MJ)	0.68	0.61	0.59	0.53

Note: ^1^ Premix A provided per kilogram of complete diet: vitamin A, 15,000 IU; vitamin D3, 1500 IU; vitamin E, 50 mg; vitamin K3, 2.5 mg; vitamin B1, 2.3 mg; vitamin B2, 4 mg; vitamin B6, 3 mg; vitamin B12, 0.03 mg; niacin, 30 mg; vitamin C, 250 mg; calcium pantothenate, 12mg; folic acid, 2 mg; biotin, 0.3 mg; Fe (FeSO_4_·H_2_O), 110 mg; Cu (CuSO_4_·5H_2_O), 15mg; I (KI), 0.50 mg; Se (Na_2_SeO_3_),0.2 mg; Zn (ZnSO_4_·H_2_O), 40 mg; Mn (MnSO_4_·H_2_O), 40 mg; Co (CoCl_2_), 0.2 mg. ^2^ Premix B provided per kilogram of complete diet: vitamin A, 12,000 IU; vitamin D3, 1500 IU; vitamin E, 50 mg; vitamin K3, 2.5 mg; vitamin B1, 2.3 mg; vitamin B2, 4 mg; vitamin B6, 3 mg; vitamin B12, 0.03 mg; niacin, 25 mg; vitamin C, 250 mg; calcium pantothenate, 10 mg; folic acid, 2 mg; biotin, 0.3 mg; Fe (FeSO_4_·H_2_O), 90 mg; Cu (CuSO_4_·5H_2_O), 12 mg; I (KI), 0.50 mg; Se (Na_2_SeO_3_), 0.2 mg; Zn (ZnSO_4_·H_2_O), 30 mg; Mn (MnSO_4_·H_2_O), 250 mg; Co (CoCl_2_), 0.2 mg. ^3^ Nutrient levels were estimated values. ^4^ HR: Diets with high standardized ileal digestible lysine to net energy ratio. ^5^ LR: Diets with low standardized ileal digestible lysine to net energy ratio.

**Table 2 animals-14-01824-t002:** Effects of dietary standardized ileal digestible lysine to net energy ratio on carcass traits and marbling score of longissimus dorsi muscle in finishing pigs.

Items	HR	LR	*p*-Value
Initial BW ^1^ (kg)	65.72 ± 0.55	64.67 ± 0.38	0.189
Pre-slaughter BW (kg)	116.85 ± 2.06	121.23 ± 2.52	0.208
Hot carcass weight (kg)	80.95 ± 2.10	85.00 ± 1.83	0.177
Dressing percentage (%)	69.49 ± 2.86	70.22 ± 1.76	0.832
Backfat thickness (mm)			
Thickest shoulders	34.78 ± 1.60	35.52 ± 1.35	0.734
Thoracolumbar junction	20.74 ± 1.63	22.68 ± 0.60	0.289
Lumbosacral junction	14.88 ± 1.69	13.46 ± 0.72	0.456
Average backfat thickness	23.47 ± 1.20	23.88 ± 0.69	0.769
Loin-eye area (cm^2^)	71.28 ± 3.31	72.40 ± 3.67	0.826
Marbling score	1.27 ± 0.14	1.94 ± 0.24 *	0.034

Note: ^1^ BW: body weight. * indicates significant difference (*p* < 0.05).

**Table 3 animals-14-01824-t003:** Effects of dietary standardized ileal digestible lysine to net energy ratio on serum biochemical indexes of finishing pigs.

Items	HR	LR	*p*-Value
GLU (mmol/L) ^1^	4.19 ± 0.28	4.62 ± 0.41	0.427
TC (mmol/L) ^2^	2.46 ± 0.18	2.26 ± 0.12	0.381
TG (mmol/L) ^3^	0.48 ± 0.02	0.66 ± 0.08	0.085
HDL-C (mmol/L) ^4^	1.16 ± 0.04	1.20 ± 0.05	0.627
LDL-C (mmol/L) ^5^	1.06 ± 0.02	1.13 ± 0.09	0.495
NEFA (mmol/L) ^6^	0.61 ± 0.04	0.60 ± 0.05	0.946

Note: ^1^ GLU: glucose. ^2^ TC: total cholesterol. ^3^ TG: triglyceride. ^4^ HDL-C: high density lipoprotein cholesterol. ^5^ LDL-C: low density lipoprotein cholesterol. ^6^ NEFA: non-esterified fatty acid.

## Data Availability

Data are contained within the article and the Appendix A.

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
