# Peer review of "Impacts of Dietary Standardized Ileal Digestible Lysine to Net Energy Ratio on Lipid Metabolism in Finishing Pigs Fed High-Wheat Diets"

_animals, 2024, doi:10.3390/ani14121824_

Round 1

Reviewer 1 Report

Comments and Suggestions for Authors

The aim of the article ‘’Impacts of dietary standardized ileal digestible lysine to net energy ratio on lipid metabolism in finishing pigs fed high-wheat diets” was to investigate the impacts of dietary standardized ileal digestible lysine to net energy (SID Lys:NE) ratio on lipid metabolism in pigs fed high-wheat diets. This is a well-designed and well-written work, providing interesting results and novel results. Therefore, I suggest the acceptance under a major revision based on the following major and minor comments. 

Abstract 

-        In the keywords place finishing pigs in the first place, it is the focus species of the study.

Introduction

-       You could elaborate on the desired organoleptic characteristic of pork and how a wheat-rich diet could provide these characteristics.

Materials and Methods

-        Provide more data on whether the pigs received any preventive treatment 

-        Why were 36 pigs selected to participate in the study? Has there been any sample size calculation?

-        Please provide further details on blood collection (e.g. restraint of the pigs, needle size)

-        Why did you select left liver lobe and colonic chyme samples? Please explain. 

Discussion:

-        Is a wheat-based diet more expensive or cheaper than a corn-based diet? Could you elaborate on the economic impact of your study on pig producers?

-        What are the negative points of the research, if any? Your recommendations for future studies, if any?

Comments on the Quality of English Language

A final editing regarding English language would be of value for the improvement of the text.

Author Response

Response to editor and reviewers' comments on animals-2950952

Dear editors & reviewers,

Thank you very much for your precious comments and suggestions about our manuscript submitted to your journal (manuscript ID: animals-2950952). Your suggestions gave us much to learn, and helped improve our scientific writing to a great extent. We have revised the manuscript according to the comments and suggestions, and the amendments are highlighted with RED front in the revised manuscript. Below you will find our point-by-point responses to your comments and questions. The whole manuscript has been carefully checked again by ourselves. We do hope we could understand your questions correctly and have given right answers in the revised manuscript. Please feel free to inform me if there are still some questions. Thank you very much in advance!

With best regards,

Wen Yao, PhD

College of Animal Science and Technology

Nanjing Agricultural University

E-mail: yaowen67jp@njau.edu.cn

Comments and Suggestions for Authors

The aim of the article "Impacts of dietary standardized ileal digestible lysine to net energy ratio on lipid metabolism in finishing pigs fed high-wheat diets" was to investigate the impacts of dietary standardized ileal digestible lysine to net energy (SID Lys:NE) ratio on lipid metabolism in pigs fed high-wheat diets. This is a well-designed and well-written work, providing interesting results and novel results. Therefore, I suggest the acceptance under a major revision based on the following major and minor comments.

Abstract

Question 1: In the keywords place finishing pigs in the first place, it is the focus species of the study.

Answer 1: Thanks for your invaluable suggestions, we have placed "finishing pigs" in the first place.

Introduction

Question 2: You could elaborate on the desired organoleptic characteristic of pork and how a wheat-rich diet could provide these characteristics.

Answer 2: Thanks for your advice. In fact, feeding high-wheat diets can decrease the intramuscular fat (IMF) content, which stands as one of the most crucial indicators for grading pork quality. The focus of this paper is to investigate the mechanism by which a high-wheat diet inhibits IMF deposition. We have modified the introduction in Line 62-71, as below: Intramuscular fat (IMF) content, represented by the marbling score, stands as one of the most crucial indicators for grading pork quality [6]. Bejerholm and Barton-Gade proposed a mini-mum of 2% IMF to ensure satisfactory eating characteristics [7], and DeVol et al reported a linear preference for IMF in terms of juiciness, tenderness and flavor range from 1.09% to 6.75% [8]. However, wheat has lower digestible energy and metabolic energy than corn when fed to barrows [9]. Feeding high-wheat diets (with wheat content at 55%) or wheat-soybean meal diets (replacing corn with wheat entirely) can compromise pork quality, such as IMF content [10, 11]. Mastering the intricate mechanism behind wheat-induced meat quality deterioration and resolving this challenge as a pivotal endeavor in the realm of scientific inquiry.

  1. Khan, R.; Raza, S. H. A.; Junjvlieke, Z.; Wang, X.; Garcia, M.; Elnour, I. E.; Wang, H.; Zan, L. Function and Transcriptional Regulation of Bovine TORC2 Gene in Adipocytes: Roles of C/EBP, XBP1, INSM1 and ZNF263. Int J Mol Sci. 2019, 20, 4338.
  2. Bejerholm, C., Barton-Gade, P. Effect of intramuscular fat level on eating quality of pig meat. Paper presented at the Proceedings of the 32nd European meeting of meat research workers. 1986.
  3. DeVol, D.; McKeith, F.; Bechtel, P. J.; Novakofski, J.; Shanks, R.; Carr, T. Variation in composition and palatability traits and relationships between muscle characteristics and palatability in a random sample of pork carcasses. J Anim Sci, 1988, 66(2), 385-395.
  4. Xie, F.; Pan, L.; Li, Z.; Shi, M.; Liu, L.; Li, Y. Huang, C.; Li, X.; Piao, X. Digestibility of energy in four cereal grains fed to barrows at four body weights. Anim Feed Sci Tech. 2017, 232, 215-221.
  5. Xia, Z.; Ye, X.; Chen, D.; Zheng, P. Effect of different dietary energy and protein sources on growth performance, carcass traits and meat quality of growing-finishing pigs. Feed Industry. 2019, 40, 31-39.
  6. Wang, J.; Xia, S.; Wang, H.; Li, H.; Zheng, W.; Yao, W. Effects of dietary standardized ileal digestible lysine to net energy ratio and wheat level on growth performance, meat quality, intestinal microbiota and body metabolism of finishing pigs. Lives Sci. 2023, 276, 105315.

Materials and Methods

Question 3: Provide more data on whether the pigs received any preventive treatment.

Answer 3: We have added the feeding management and immunization program information in Line 97-98, as below: The feeding management and immunization program were conducted according to the general requirements of the pig farm.

Question 4: Why were 36 pigs selected to participate in the study? Has there been any sample size calculation?

Answer 4: The 36 pigs were determined was determined by referencing previous studies.

  1. Chen, C.; Zhou, Y.; Fu, H.; Xiong, X.; Fang, S.; Jiang, H.; Wu, J.; Yang, H.; Gao J.; Huang, Expanded catalog of microbial genes and metagenome-assembled genomes from the pig gut microbiome. Nat Commun. 2021, 12(1), 1106.
  2. Yang, Y.; Zhao, X.; Le, M. H.; Zijlstra, R. T.; Gänzle, M. Reutericyclin producing Lactobacillus reuteri modulates development of fecal microbiota in weanling pigs. Front Microbiol. 2015, 28(6), 762.

Question 5: Please provide further details on blood collection (e.g. restraint of the pigs, needle size)

Answer 5: We have provided the information in Line 119-121, as below: Pigs were restrained by snout snare, and blood samples were collected by using a 14-gauge needle via jugular venipuncture from anterior vena cave.

Question 6: Why did you select left liver lobe and colonic chyme samples? Please explain.

Answer 6: The liver sample collection method was determined by referencing a previous study [1].

Intestinal microorganism and their metabolic products, short-chain fatty acids (SCFAs), play a significant role in intramuscular fat deposition process. SCFAs were produced when indigestible carbohydrates (like dietary fiber and resistant starch) enter the large intestine and are fermented by microflora. Colon is the major site of SCFAs production [2], and 95% of SCFAs are absorbed from the colon [3,4]. Infusions of SCFAs in the colon affected lipid and energy metabolism in volunteers who were overweight or obese [5,6]. Thus, colonic chyme samples were selected.

  1. Yang, Z.; Wang, F.; Yin, Y.; Huang, P.; Jiang, Q.; Liu, Z.; Yin, Y.; Chen, J. Dietary Litsea cubeba essential oil supplementation improves growth performance and intestinal health of weaned piglets. Anim Nutr. 2023, 13, 9-18.
  2. Ruppin, H.; Bar-Meir, S.; Soergel, K. H.; Wood, C. M.; & Schmitt Jr, M. G. Absorption of short-chain fatty acids by the colon. Gastroenterology, 1980, 78(6), 1500-1507.
  3. Sakata, T. Pitfalls in short‐chain fatty acid research: a methodological review. Animal Science Journal, 2019, 90(1), 3-13.
  4. Uehara, M.; Inoue, T.; Kominato, M.; Hase, S.; Sasaki, E.; . . . Sakakibara, Y. Intraintestinal Analysis of the Functional Activity of Microbiomes and Its Application to the Common Marmoset Intestine. mSystems, 2022, 5, e0052022.
  5. Canfora, E. E.; van der Beek, C. M.; Jocken, J. W.; Goossens, G. H.; Holst, J. J.; Olde Damink, S. W.; ... & Blaak, E. E. Colonic infusions of short-chain fatty acid mixtures promote energy metabolism in overweight/obese men: a randomized crossover trial. Scientific reports, 2017, 7(1), 2360.
  6. van der Beek, C. M.; Canfora, E. E.; Lenaerts, K.; Troost, F. J.; Olde Damink, S. W.; Holst, J. J.; ... & Blaak, E. E. Distal, not proximal, colonic acetate infusions promote fat oxidation and improve metabolic markers in overweight/obese men. Clinical Science, 2016, 130(22), 2073-2082.

Discussion:

Question 7: Is a wheat-based diet more expensive or cheaper than a corn-based diet?

Answer 7: As shown in the table below, wheat-based diets are cheaper than corn-based diets at each phase when nutrient levels were equal.

W55HR1

W55LR2

W0HR3

W0LR4

65-90 kg (¥/t)

3103

3129

¥3354

3338

90-110 kg (¥/t)

3163

3124

3132

3148

Note: 1 W55HR: Wheat-based diet with high SID Lys:NE ratio. 2 W55LR: Wheat-based diet with low SID Lys:NE ratio. 3 W0HR: Corn-based diet with high SID Lys:NE ratio. 4 W0LR: Corn-based diet with low SID Lys:NE ratio.

Question 8: Could you elaborate on the economic impact of your study on pig producers?

Answer 8: Using wheat as an energy component in pig feed will effectively reduce the utilization of corn and soybean meal, thereby lowering feed costs. However, feeding high-wheat diets (with wheat content at 55%) or wheat-soybean meal diets (replacing corn with wheat entirely) can compromise pork quality. Our study indicated that the decrease in meat quality caused by excessive wheat levels in the diet can be improved by reducing dietary SID Lys:NE ratio. This study contributes to promoting the use of wheat in pig diets, reducing feed costs, and increasing farmers' income. Additionally, although the improvement in meat quality cannot be quantified by price, pigs with higher meat quality are undoubtedly more popular, thus producers have greater bargaining power.

Question 9: What are the negative points of the research, if any? Your recommendations for future studies, if any?

Answer 9: In my opinion, the only negative point is that the number of replicates during feeding phase is slightly low. For future studies, we should increase the number of replicates and the size of the trial pig population.

Reviewer 2 Report

Comments and Suggestions for Authors

Submitted for review is a manuscript entitled: Impacts of dietary standardized ileal digestible lysine to net energy ratio on lipid metabolism in finishing pigs fed high-wheat diets. The authors have not avoided some errors and shortcomings. I have indicated my suggestions or questions/concerns below:

Note 1 – Line 83-93 – Section Materials and Methods, subsection Animal and experimental design: In this subsection, the authors should add information on the environmental conditions in which the animals were reared and the ethical committee's relevant approvals.

Note 2 – Line 108-110 – Section Materials and Methods, subsection Sample collection: In this subsection, the authors should indicate the range of animal body weights (minimum-maximum)

Note 3 – Line 204-211 – Section Results, subsection Carcass traits and marbling score of LD: The authors use abbreviations in the description of the results, while the table contains the whole names of the parameters. I recommend standardising the notation so that it is fully understandable to the reader. Perhaps it would be worth considering creating a separate subsection for the abbreviations used together with their full names and using only the relevant abbreviations in the text of the manuscript. Furthermore, in the phrase: „And dietary SID Lys:NE ratio had no significant effects on HCW, dressing percentage, backfat thickness, and LEA (P > 0.05)” only the no significant effect against some parameters was indicated. This raises the question of what about the other parameters studied. I recommend that the authors once again examine the description of the results in detail throughout the manuscript.

The manuscript submitted for evaluation can be published after taking into account the above-mentioned revisions.

Author Response

Response to editor and reviewers' comments on animals-2950952

Dear editors & reviewers,

Thank you very much for your precious comments and suggestions about our manuscript submitted to your journal (manuscript ID: animals-2950952). Your suggestions gave us much to learn, and helped improve our scientific writing to a great extent. We have revised the manuscript according to the comments and suggestions, and the amendments are highlighted with RED front in the revised manuscript. Below you will find our point-by-point responses to your comments and questions. The whole manuscript has been carefully checked again by ourselves. We do hope we could understand your questions correctly and have given right answers in the revised manuscript. Please feel free to inform me if there are still some questions. Thank you very much in advance!

With best regards,

Wen Yao, PhD

College of Animal Science and Technology

Nanjing Agricultural University

E-mail: yaowen67jp@njau.edu.cn

Comments and Suggestions for Authors

Submitted for review is a manuscript entitled: Impacts of dietary standardized ileal digestible lysine to net energy ratio on lipid metabolism in finishing pigs fed high-wheat diets. The authors have not avoided some errors and shortcomings. I have indicated my suggestions or questions/concerns below:

Question 1: Line 83-93 – Section Materials and Methods, subsection Animal and experimental design: In this subsection, the authors should add information on the environmental conditions in which the animals were reared and the ethical committee's relevant approvals.

Answer 1: We have added the information on the environmental conditions in Line 95-97, as blow: Pigs were housed in an environmentally controlled building with pens equipped with a feeder, a nipple drinker, and slotted floors during the experiment period.

The ethical committee's relevant approvals were illustrated in Line 100-101, as below: The experiment was approved by the Animal Care and Use Committee of Nanjing Agricultural University (Approve number SYXK (Su) 2011-0036).

Question 2: Line 108-110 – Section Materials and Methods, subsection Sample collection: In this subsection, the authors should indicate the range of animal body weights (minimum-maximum)

Answer 2: We have revised the description of the pig selection method and indicated the range of body weights in Line 117-118, as below: When body weight (BW) reached market weight, twelve pigs with BW ranging 107.04 to 127.10 kg (two pigs with medium BW from each replicate) were chosen for evaluation.

Question 3: Line 204-211 – Section Results, subsection Carcass traits and marbling score of LD: The authors use abbreviations in the description of the results, while the table contains the whole names of the parameters. I recommend standardising the notation so that it is fully understandable to the reader. Perhaps it would be worth considering creating a separate subsection for the abbreviations used together with their full names and using only the relevant abbreviations in the text of the manuscript. Furthermore, in the phrase: „And dietary SID Lys:NE ratio had no significant effects on HCW, dressing percentage, backfat thickness, and LEA (P > 0.05)” only the no significant effect against some parameters was indicated. This raises the question of what about the other parameters studied. I recommend that the authors once again examine the description of the results in detail throughout the manuscript.

Answer 3: Thanks for your suggestion. We have defined the abbreviation (LD) at first mention in Line 127.

We have rewritten the effects of dietary SID Lys:NE ratio on carcass traits and marbling score of LD in Line 215-220, as below: The effects of dietary treatment on carcass traits and marbling score of LD were presented in Table 2. The marbling score of LD was increased in pigs fed with LR diets compared to HR diets (P < 0.05). And dietary SID Lys:NE ratio had no significant effects on pre-slaughter BW, HCW, dressing percentage, backfat thickness at thickest shoulders, thoracolumbar junction, lumbosacral junction, and average backfat thickness, as well as LEA (P > 0.05).

Additionally, we have examined the description of the results carefully. Thank you very much!

Reviewer 3 Report

Comments and Suggestions for Authors

Comment:

Significant research that showed us changes in pig nutrition to impacts on carcass trait and regulation in lipid metabolism in finishing pigs fed diets with low standardized ileal digestible lysine to net energy ratio.

The main question addressed by the research is, the possibility of reducing the share of corn with a higher share of wheat and the impact of this on the quality of meat and meat, as well as on lipid status - improved energy metabolism, stimulated lipogenesis and inhibited lipolysis.  

The paper explains in one part the lipid mechanism - feeding high wheat diets or wheat soy bean meal diets (replacing corn) can compromise pork quality, such as intramuscular fat (IMF) content which is extremely important for the quality of the meat and the demands of consumers (marbling score etc.).  

Compared to other published work by far, the results indicated that pigs had higher expression levels of lipogenic genes (FABP4 and ACCα) and a lower expression level of ATGL in the LR group than in the HR group, suggesting that diminishing dietary SID Lys:NE ratio triggered increased lipogenesis and inhibited lipolysis. Meanwhile, the expression level of AMPKα1, a key regulator of lipid metabolism through multiple mechanisms, was downregulated in the LR group. The downregulation of AMPKα1, SIRT1 and PGC-1α in the LR group hinted at the possibility that diminishing dietary SID Lys:NE ratio improved energy metabolism. More energy was used for lipid synthesis, and energy production from fatty acids was inhibited. On the one hand, they proved that AMPKα1 directly regulated the expression of downstream lipid metabolism genes.    

Regarding the methodology: Authors should consider whether and to what extent it affects the intestinal microbiota and the health status of pigs.

The conclusions are consistent with the evidence and arguments presented.

All references are appropriate and relevant to the given topic and discussion.

The quality of the data is excellent and the tables and figures are clearly presented with a good and clear legend.

Author Response

Response to editor and reviewers' comments on animals-2950952

Dear editors & reviewers,

Thank you very much for your precious comments and suggestions about our manuscript submitted to your journal (manuscript ID: animals-2950952). Your suggestions gave us much to learn, and helped improve our scientific writing to a great extent. We have revised the manuscript according to the comments and suggestions, and the amendments are highlighted with RED front in the revised manuscript. Below you will find our point-by-point responses to your comments and questions. The whole manuscript has been carefully checked again by ourselves. We do hope we could understand your questions correctly and have given right answers in the revised manuscript. Please feel free to inform me if there are still some questions. Thank you very much in advance!

With best regards,

Wen Yao, PhD

College of Animal Science and Technology

Nanjing Agricultural University

E-mail: yaowen67jp@njau.edu.cn

Question 1: Regarding the methodology: Authors should consider whether and to what extent it affects the intestinal microbiota and the health status of pigs.

Answer 1: Thanks for your suggestion. In the present study, dietary SID Lys:NE ratio had no significant effects on α- and b-diversity. Reducing dietary SID Lys:NE ratio significantly increased the relative abundances of YRC22, Parabacteroides, Sphaerochaeta, and Bacteroides, whereas decreased the proportions of Roseburia, f_Lachnospiraceae_g_Clostridium, Enterococcus, Shuttleworthia, Exiguobac-terium, Corynebacterium, Subdoligranulum, Sulfurospirillum, and Marinobacter. The concentration of serum glucose, total cholesterol, triglyceride, high density lipoprotein cholesterol, low density lipoprotein cholesterol, and non-esterified fatty acid in two group were within the normal range, this indicated that diminishing dietary SID Lys:NE ratio had no adverse effects on the health status of pigs to some extent. Thanks for your advice again, we will measure appropriate indicators to reflect the health status of the pigs in future studies.

Reviewer 4 Report

Comments and Suggestions for Authors

In general, the manuscript is original and well-written. The paper provides important insights into how SID Lys to NE ratios can influence energy metabolism and their effects on fat tissue development. However, there are a few limitations. For example, performance variables and important pork traits (such as IMF, cooking losses, shear force, and drip loss) were not analyzed. The most concerning factor is that there were only three replications for each treatment, which could explain the lack of statistical differences between some variables, especially for carcass traits. Therefore, this could be considered a confounding factor. Descriptive analysis might be a more suitable option (particularly for carcass traits), as it allows for a comprehensive overview of the data, highlighting key patterns and trends without the constraints of inferential statistics.

Abstract

Line 21-22: Please provide more details, such as initial body weight, number of replicates, and animals per pen.

Line 42: I suggest replacing the word “diminishing” with “reduced dietary SID Lys:NE ratio improves…”

Line 44-45: The meaning of the sentence “Whereas colonic microbial composition changes were responsible to the body metabolism alterations” is unclear. I suggest rewriting it.

Throughout the abstract, I recommend removing the word “significantly.”

Introduction

Line 74-76: Please cite the sentence: "Host low energy states and colonic short-chain fatty acids (SCFAs)-producing bacteria alteration were responsible for the reduction in IMF content."

Materials and Methods

Line 92: In my opinion, the number of replicates is too low and may result in Type II error. Therefore, the lack of statistical differences between some variables, especially for carcass traits, could be explained by the low number of replicates. For example, there was a notable numerical difference between treatments of 5kg and 5kg for final BW and carcass weight, respectively.

Table 1: I recommend reorganizing the diet columns, first by phase and then by treatment. Regarding the nutritional values, many nutritionists use digestible energy instead of net energy during feed formulation. Therefore, I think it is appropriate to show the digestible energy and its ratio to SID Lys in the table as well.

Line 123: Please define LD (Longissimus dorsi) in the manuscript before using the abbreviation.

Line 123: Is there any particular reason why some important pork traits, such as shear force, intramuscular fat content, drip loss, and cooking losses, were not analyzed?

Line 128: Please correct the word “lion” to “loin.”

Results

I suggest removing the word “significantly” throughout the entire section.

Table 2: Were performance variables measured? Additionally, I recommend presenting the initial body weight for each treatment.

Line 146: The results of (PPARα: peroxisome proliferator-activated receptor α) were not presented in Figure 1. Therefore, PPARα should be removed from the sentence.

Line 250-251: The sentence “Data are expressed as the means ± SEM” must be corrected since the mean and SEM values were not presented in the figures. The same observation applies to Figures 3 and 4.

Line 315: In Figure 5B, the color of the dots representing downregulated metabolites is “blue” and not “green.”

Discussion

Line 331-337: As reducing the SID Lys to NE ratio increased marbling score, is there any mechanism that may explain the lack of difference in backfat thickness?

Line 345: Please briefly describe the main role of the gene “ATGL.”

Line 356-357: In the sentence “the lower expression level in the LR group,” it must be clear that there was a “numerical difference.”

Line 363: The importance of “CYP7B1” in bile acids biosynthesis should be explained.

Comments on the Quality of English Language

The text is well-written and adheres to English language standards, demonstrating clarity, coherence, and proper grammatical structure throughout.

Author Response

Response to editor and reviewers' comments on animals-2950952

Dear editors & reviewers,

Thank you very much for your precious comments and suggestions about our manuscript submitted to your journal (manuscript ID: animals-2950952). Your suggestions gave us much to learn, and helped improve our scientific writing to a great extent. We have revised the manuscript according to the comments and suggestions, and the amendments are highlighted with RED front in the revised manuscript. Below you will find our point-by-point responses to your comments and questions. The whole manuscript has been carefully checked again by ourselves. We do hope we could understand your questions correctly and have given right answers in the revised manuscript. Please feel free to inform me if there are still some questions. Thank you very much in advance!

With best regards,

Wen Yao, PhD

College of Animal Science and Technology

Nanjing Agricultural University

E-mail: yaowen67jp@njau.edu.cn

Comments and Suggestions for Authors

In general, the manuscript is original and well-written. The paper provides important insights into how SID Lys to NE ratios can influence energy metabolism and their effects on fat tissue development. However, there are a few limitations. For example, performance variables and important pork traits (such as IMF, cooking losses, shear force, and drip loss) were not analyzed. The most concerning factor is that there were only three replications for each treatment, which could explain the lack of statistical differences between some variables, especially for carcass traits. Therefore, this could be considered a confounding factor. Descriptive analysis might be a more suitable option (particularly for carcass traits), as it allows for a comprehensive overview of the data, highlighting key patterns and trends without the constraints of inferential statistics.

Abstract

Question 1: Line 21-22: Please provide more details, such as initial body weight, number of replicates, and animals per pen.

Answer 1: Thanks for your suggestion. We have modified the sentence in Line 89-92, as below: Thirty-six crossbred (Duroc ´ Landrace ´ Yorkshire) growing barrows (65.20 ± 0.38 kg) were blocked into two treatment groups, fed high-wheat diets with either high SID Lys:NE ratio (HR) or low SID Lys:NE ratio (LR). Each treatment group consisted of three replicates, with six pigs per pen in each replicate.

Question 2: Line 42: I suggest replacing the word “diminishing” with “reduced dietary SID Lys:NE ratio improves…”

Answer 2: According to your advice, we have revised the sentence in Line 37-38, as below: Reduced dietary SID Lys:NE ratio improves energy metabolism, stimulated lipogenesis and in-hibited lipolysis in finishing pigs by regulating AMPKα/SIRT1/PGC-1α pathway and FXR/SHP pathway.

Question 3: Line 44-45: The meaning of the sentence “Whereas colonic microbial composition changes were responsible to the body metabolism alterations” is unclear. I suggest rewriting it.

Answer 3: We have rewritten the sentence in Line 39-41, as below: Whereas Parabacteroides and Sphaerochaeta benefited BAs synthesis, whereas f_Lachnospiraceae_g_Clostridium, Shuttleworthia and Marinobacter may contribute to the activation of AMPK signaling pathway.

Question 4: Throughout the abstract, I recommend removing the word “significantly.”

Answer 4: We have deleted “significantly” in the abstract.

Introduction

Question 5: Line 74-76: Please cite the sentence: "Host low energy states and colonic short-chain fatty acids (SCFAs)-producing bacteria alteration were responsible for the reduction in IMF content."

Answer 5: We have added the references for the sentence.

  1. Bäckhed, F.; Ding, H.; Wang, T.; Hooper, L. V.; Koh, G. Y.; Nagy, A.; Semenkovich, C. F.; Gordon, J. I. The gut microbiota as an environmental factor that regulates fat storage. Proc Natl Acad Sci U S A, 2004, 101(44), 15718-15723.

Materials and Methods

Question 6: Line 92: In my opinion, the number of replicates is too low and may result in Type II error. Therefore, the lack of statistical differences between some variables, especially for carcass traits, could be explained by the low number of replicates. For example, there was a notable numerical difference between treatments of 5kg and 5kg for final BW and carcass weight, respectively.

Answer 6: Thanks for your recommendation. Although we set up three replicates for each treatment group, six pigs from each treatment (two per replicate with medium BW) were chosen for evaluation at the end of experiment. Thus, the six pig per group were used to evaluate the effect of dietary SID Lys:NE ratio on the carcass traits, marbling score of longissimus dorsi muscle, serum biochemical indexes, gene expression, microbial community, and other indicators studied in this research. And according to previous studies on pigs, having 6 pig per group is adequately substantial to attain statistically significant results, with the sample size being ample to ensure result representativeness. The references are presented as follows:

  1. Li, S.; Wang, H.; Wang, X.; Wang, Y.; Feng, J. Betaine affects muscle lipid metabolism via regulating the fatty acid uptake and oxidation in finishing pig. J Anim Sci Biotechnol, 2017, 8(1), 1-9.
  2. Ma,; Zhang, J.; Wang, Z.; Yu, Q.; Han, L. Effects of muscle-specific oxidative stress on protein phosphorylation and its relationship with mitochondrial dysfunction, muscle oxidation, and apoptosis. Food Chem. 2023, 427, 136737.

Question 7: Table 1: I recommend reorganizing the diet columns, first by phase and then by treatment. Regarding the nutritional values, many nutritionists use digestible energy instead of net energy during feed formulation. Therefore, I think it is appropriate to show the digestible energy and its ratio to SID Lys in the table as well.

Answer 7: Thanks for your suggestion, we have adjusted the table and the digestible energy and its ratio to SID Lys was added.

Items

65-90 kg

90-110 kg

HR 4

LR 5

HR

LR

Ingredient, g

Wheat

550.00

550.00

550.00

550.00

Corn

129.40

160.40

168.00

193.00

Soybean meal

110.00

75.00

54.00

33.00

Wheat middings

100.00

100.00

200.00

200.00

Rice bran

75.00

75.00

Limestone

11.00

8.60

10.00

8.50

Soybean oil

8.00

L-Lysine·HCl (70%)

5.80

5.45

5.20

4.50

DL-Methionine

0.95

0.65

0.55

0.20

L-Tryptophan

0.40

0.30

L-Threonine

1.80

1.60

1.60

1.30

Valine

0.35

0.25

NaCl

3.80

2.70

3.70

2.70

CaHCO3

3.70

4.20

NaHCO3

2.10

2.40

1.90

1.70

Choline chloride

0.50

0.50

0.20

0.20

CuSO4

0.30

0.30

0.30

0.30

Premix A 1

10.00

10.00

Premix B 2

10.00

10.00

Total

1005.10

1005.35

1005.45

1005.40

Nutrient composition 3, %

Net energy (NE, MJ/kg)

9.75

10.08

9.62

9.96

Digestible energy (DE, MJ/kg)

13.73

13.95

13.61

13.63

Crude protein

16.07

14.74

14.44

13.63

Crude fat

3.01

3.88

2.33

2.39

Crude fiber

3.19

3.05

2.93

2.86

Standardized ileal digestible amino acids, %

Lysine (Lys)

0.94

0.85

0.80

0.72

Methionine + Cysteine

0.56

0.50

0.49

0.44

Threonine

0.61

0.55

0.54

0.48

Tryptophan

0.18

0.16

0.15

0.14

Lys:NE (g/MJ)

0.96

0.84

0.83

0.72

Lys:DE (g/MJ)

0.68

0.61

0.59

0.53

Question 8: Line 123: Please define LD (Longissimus dorsi) in the manuscript before using the abbreviation.

Answer 8: Thanks for your advice. We have defined the abbreviation (LD) at first mention in the text (Line 127).

Question 9: Line 123: Is there any particular reason why some important pork traits, such as shear force, intramuscular fat content, drip loss, and cooking losses, were not analyzed?

Answer 9: Dear reviewer, we completed the fattening of pigs in a commercial pig farm. The pork traits you mentioned, such as shear force, drip loss, and so on, should be analyzed with fresh pork sample. Unfortunately, we lacked the manpower and appropriate instruments to conduct the aforementioned measurements. Your suggestion reminds us that more important pork traits should be measured in future studies. Thank you very much!

Question 10: Line 128: Please correct the word "lion" to "loin."

Answer 10: Thanks for your reminder. We have replaced the "lion" with "loin".

Results

Question 11: I suggest removing the word “significantly” throughout the entire section.

Answer 11: We have deleted “significantly” in the Results section.

Question 12: Table 2: Were performance variables measured? Additionally, I recommend presenting the initial body weight for each treatment.

Answer 12: As seen in the table below, dietary SID Lys:NE ratio had no significant effect on the growth performance of finishing pigs. The growth performance data was presented in the supplemental Material. And we supplemented the discussion about growth performance in Line 336-338, as below: The growth performance and carcass yield of finishing pigs directly influences the breeders' profits from production of pork. In our research, dietary SID Lys:NE ratio showed no significant effects on growth performance of finishing pigs. Meanwhile, the initial body weight data was added in the Table 2.

Table S1. Effects of dietary standardized ileal digestible lysine to net energy ratio on growth performance of finishing pigs.

Items

HR

LR

P-value

65-90 kg

ADG, kg 1

1.13±0.05

1.11±0.04

0.697

ADFI, kg 2

3.31±0.15

3.11±0.06

0.310

F:G 3

2.922±0.038

2.820±0.084

0.316

90-110 kg

ADG, kg

1.09±0.05

1.03±0.03

0.355

ADFI, kg

3.58±0.15

3.21±0.06

0.088

F:G

3.280±0.008

3.113±0.084

0.229

65-110 kg

ADG, kg

1.11±0.05

1.08±0.02

0.506

ADFI, kg

3.42±0.15

3.15±0.05

0.167

F:G

3.067±0.023

2.931±0.063

0.112

Note: 1 ADG: average daily gain. 2 ADFI: average daily feed intake. 3 F:G: the ratio of feed to gain.

Question 13: Line 146: The results of (PPARα: peroxisome proliferator-activated receptor α) were not presented in Figure 1. Therefore, PPARα should be removed from the sentence.

Answer 13: Thanks for your careful review. We have removed " PPARα" throughout entire text.

Question 14: Line 250-251: The sentence “Data are expressed as the means ± SEM” must be corrected since the mean and SEM values were not presented in the figures. The same observation applies to Figures 3 and 4.

Answer 14: We have corrected the sentence in Line 205-206, as below: Data (carcass traits, marbling score of LD, serum biochemical indexes, genes expression levels) were expressed as the means and their SEM.

Question 15: Line 315: In Figure 5B, the color of the dots representing downregulated metabolites is "blue" and not "green."

Answer 15: We have replaced "green" with "blue".

Discussion

Question 16: Line 331-337: As reducing the SID Lys to NE ratio increased marbling score, is there any mechanism that may explain the lack of difference in backfat thickness?

Answer 16: Increasing dietary energy density promoted intramuscular fat (IMF) deposition via enhancing the expression levels of peroxisome proliferator-activated receptor γ [1], acetyl CoA carboxylase, fatty acid synthase, and fatty acid binding protein (FABP) [2]. And High energy diets with a reduced lysine: energy ratio increased IMF content, while appropriately reducing the Lys: energy ratio inhibited the high energy diet-induced increase in backfat thickness and total fat in pigs [3]. Thus, we think the effects of dietary SID Lys:NE ratio on the lipid metabolism of pigs is tissue-specific. This study aimed to study the impacts of dietary SID Lys:NE ratio on the body metabolism and intestinal microbiota, as well as its correlation with intramuscular fat deposition. The mechanisms by which dietary SID Lys:NE ratio affects backfat thickness require further investigation.

  1. Zou, T.; Mao, X.; Yu, B.; He, J.; Zheng, P.; Yu, J.; Chen, D. Effects of dietary energy density and apparent ileal digestible lysine: digestible energy ratio on growth performance, meat quality, and peroxisome proliferator-activated receptor γ (PPARγ) gene expression of muscle and adipose tissues in Landrace× Rongchang crossbred pigs. Livest Sci. 2014, 167, 219–226.
  2. Chen, J.; Chen, F.; Lin, X.; Wang, Y.; He, J.; Zhao, Y. Effect of excessive or restrictive energy on growth performance, meat quality, and intramuscular fat deposition in finishing Ningxiang pigs. Animals, 2020, 11 (1), 27.
  3. Liu, Y.; Kong, X.; Jiang, G.; Tan, B.; Deng, J.; Yang, X.; Li, F.; Xiong, X.; Yin, Y. Effects of dietary protein/energy ratio on growth performance, carcass trait, meat quality, and plasma metabolites in pigs of different genotypes. J Anim Sci Biotechnol, 2015, 6 (1), 1–10.

Question 17: Line 345: Please briefly describe the main role of the gene “ATGL.”

Answer 17: We have added the sentence in Line 361-362, as below: ATGL is a key regulatory enzyme involved in lipolysis and participates in triglyceride-specific.

Question 18: Line 356-357: In the sentence “the lower expression level in the LR group,” it must be clear that there was a “numerical difference.”

Answer 18: Thanks for your advice. We have modified the sentence in Line 365-366, as below: we observed a numerical decrease in CPT expression level in the LR group, which indicated that LR diets inhibited fatty acids β-oxidation.

Question 19: Line 363: The importance of “CYP7B1” in bile acids biosynthesis should be explained.

Answer 19: We have illustrated the role of CYP7B1 in Line 373-375, as below: In the alternative pathway, cholesterol is catalyzed by CYP27A1 to produce 27-hydroxy cholesterol, which is further catalyzed by CYP7B1 to produce chenodeoxycholic acid.

Round 2

Reviewer 1 Report

Comments and Suggestions for Authors

The authors have sufficiently addressed my comments. Therefore, I recommend the acceptance of their paper for publication in Animals.

Author Response

We have sufficiently addressed reviewer's comments and no questions need to be answered.